# Octopamine signaling from clock neurons plays dual roles in *Drosophila* long-term memory

**Yuto Kurata**[1], **Taishi Yoshii**[2], **Takaomi Sakai**[1]*

**1** Department of Biological Sciences, Tokyo Metropolitan University, Tokyo, Japan, **2** Graduate School of Environmental, Life, Natural Science and Technology, Okayama University, Okayama, Japan

* sakai-takaomi@tmu.ac.jp

## Abstract

Circadian clock genes are best known for regulating circadian rhythms, but they also play crucial roles in memory processes. This suggests that memory is modulated by neural networks containing clock neurons, although the underlying mechanisms remain unclear. In *Drosophila melanogaster*, approximately 240 clock neurons are grouped into at least eight distinct clusters. Among them, the dorsal–lateral neurons (LNds) are required for maintaining long-term memory (LTM). In contrast, the neuropeptide Pigment-dispersing factor (Pdf), expressed in both small and large ventral–lateral neurons (s-LNvs and l-LNvs, respectively), functions as a circadian output signal and is also essential for maintaining LTM. In addition, Pdf-expressing neurons (hereafter, Pdf neurons) release neurotransmitters other than Pdf, which are involved in LTM consolidation. However, the specific transmitters used by LNds and Pdf neurons in LTM processing have remained unknown. Here, we show that octopamine signaling from LNds is essential for LTM maintenance, whereas octopamine in Pdf neurons is essential for LTM consolidation. Temporally restricted knockdown of *Tyramine β hydroxylase* (*Tbh*), the gene encoding the enzyme required for octopamine synthesis, disrupted LTM maintenance when targeted in LNds, whereas it impaired LTM consolidation when targeted in Pdf neurons. Notably, *Tbh* knockdown in LNds or Pdf neurons had minimal effects on circadian behavioral rhythms or sleep. These findings reveal that octopamine released from specific subtypes of clock neurons independently regulates distinct phases of LTM in *Drosophila*.

## Author summary

Animal memory formed through learning is stabilized by a process called consolidation and becomes long-term memory (LTM), which can persist for extended periods. The fruit fly *Drosophila melanogaster* is widely used in memory research, enabling the discovery of key genes and signaling pathways. In addition, *Drosophila* has contributed significantly to circadian rhythm research through

**Data availability statement:** All data are in the manuscript and/or supporting information files.

**Funding:** This work was supported by JSPS KAKENHI (Grant number 21H02528 to T.S) and a Grant-in-Aid for Scientific Research on Innovative Areas (Singularity Biology) from the Ministry of Education, Culture, Sports, Science, and Technology of Japan (Grant number 21H00434 to T.S). The funders had no role in study design, data collection and analysis, decision to publish, or preparation of the manuscript.

**Competing interests:** The authors have declared that no competing interests exist.

studies of clock neurons and clock genes. Notably, clock neurons have been implicated in LTM regulation, suggesting a potential link between circadian clock neurons and memory function. However, the neural network connecting these processes remains poorly understood. Identifying the neurotransmitters released from clock neurons is critical to clarifying how these neurons affect memory function. In this study, we show that octopamine, the fly equivalent of the vertebrate neurotransmitter norepinephrine, released from two distinct groups of clock neurons regulates different phases of LTM in *Drosophila*: one group supports consolidation, whereas the other supports maintenance. Our findings demonstrate that clock neurons play an active role in memory regulation, and they provide new insights into the neural circuitry that controls LTM.

## Introduction

Animals acquire memories through learning, and these memories support their survival. Newly acquired memories are initially unstable, but under certain conditions, they can become stabilized in the brain—a process known as consolidation—resulting in long-term memory (LTM). Once consolidated, LTM can persist in the brain for extended periods. The fruit fly *Drosophila melanogaster* serves as a powerful model for uncovering the molecular mechanisms underlying memory, owing to its rich genetic toolkit and extensive collection of mutant stocks. Courtship conditioning, one of the memory assays in *Drosophila,* has been used in studies to reveal the molecular and cellular bases of memory [1–4]. In this assay, virgin males are paired with mated females that emit aversive stimuli—such as inhibitory pheromones and physical rejection—leading to reduced male courtship behavior. This courtship suppression is based on past memory (hereafter referred to as "courtship memory"), as shown by the observation that mutant males with memory defects fail to show this response [2,3]. A 1 h conditioning period induces courtship suppression lasting up to 8 h, but not beyond one day [5]. In contrast, 7 h conditioning induces courtship suppression that persists for at least 5 days [6]. Thus, 1 h conditioning is commonly used to evaluate short-term memory (STM), whereas 7 h conditioning is used to assess LTM [7].

Genetic studies of courtship LTM have pointed out that the molecular and cellular mechanisms underlying memory one day after conditioning differ from those two days after conditioning [3,7]. Therefore, the period up to the day after conditioning has been conceptually termed the LTM consolidation phase, whereas the period from the second day onward has been termed the LTM maintenance phase [3,7]. The same terminologies are used in this paper. Many genes required for consolidating and maintaining courtship LTM have been identified [2,3]. Many of these genes are expressed in the mushroom body (MB), a key brain structure involved in various forms of memory, highlighting its central role in regulating courtship LTM [3]. Furthermore, several LTM genes are not expressed in the MB but rather in clock neurons, which also express circadian clock genes [3,5,6,8,9]. This suggests that courtship

LTM is orchestrated by a complex neural network that integrates both MB and clock neuron functions, although the precise mechanisms remain unclear. The adult *Drosophila* brain contains about 240 clock neurons, organized into at least eight clusters, namely, the small and large ventral–lateral neurons (s-LNvs and l-LNvs), the 5th s-LNv, the dorsal–lateral neurons (LNds), lateral–posterior neurons (LPNs), and three types of dorsal neuron (DN1, DN2, and DN3) [10–13]. We previously identified a subset of LNds relevant to regulating courtship LTM [9]. In *Drosophila*, the thermogenetic tool *shibire*^ts1 (*shi*^ts1), which encodes a temperature-sensitive mutation in a Dynamin protein [14], is commonly used to transiently block neurotransmission. At the permissive temperature (PT, 25°C), synaptic transmission proceeds normally. However, shifting to a restrictive temperature (RT, 30°C) inhibits synaptic vesicle recycling, thereby disrupting neurotransmission in the targeted neurons [14]. When *shi*^ts1 is expressed in a subset of LNds, blocking neurotransmission during the LTM maintenance phase—but not during consolidation or recall—impairs courtship LTM [9]. These findings suggest that neurotransmitter release from a subset of LNds is critical for maintaining courtship LTM (Fig 1A). However, the specific neurotransmitters released from LNds essential for LTM maintenance remain unidentified.

In *Drosophila*, environmental light is essential for the maintenance of courtship LTM, and the Pigment-dispersing factor (Pdf), known for its role in circadian behavioral rhythms [15] and light-induced arousal [16,17], also contributes to light-dependent LTM maintenance [8]. Pdf-expressing clock neurons (hereafter, Pdf neurons), which include s-LNvs and l-LNvs, directly detect light through visual systems [18] and circadian photoreceptors [19,20]. Light exposure triggers Pdf secretion from l-LNvs, supporting LTM maintenance under light conditions [21]. Blocking neurotransmission in Pdf neurons using Shi^ts1 has minimal effects on circadian behavioral rhythms, indicating that Pdf secretion is likely unaffected by Shi^ts1 [22]. Nevertheless, the blocking neurotransmission in Pdf neurons impairs LTM consolidation [5]. Therefore, it is likely that Pdf neurons secrete a neurotransmitter that is distinct from Pdf and is essential for LTM consolidation [3]. Taken together, Pdf neurons play vital roles in regulating LTM processes in a Pdf-dependent and Pdf-independent manner (Fig 1B). However, the non-Pdf neurotransmitters released from Pdf neurons that are necessary for LTM consolidation remain to be identified (Fig 1B).

How do LNds and Pdf neurons—both anatomically distant from MB neurons—regulate MB-dependent memory functions? As a first step toward uncovering intercellular communication between clock neurons and MB neurons, it is essential to identify which neurotransmitters are released from clock neurons. RNA-seq analysis of *Drosophila* clock neurons (GEO accession: GSE77451) reveals the expression of several neurotransmitter-related genes in LNds, including *Tyrosine decarboxylase 2* (*Tdc2*), *Tyramine β hydroxylase* (*Tbh*), *Choline acetyltransferase* (*ChAT*), and *Vesicular acetylcholine transporter* (*VAChT*) [23]. Tdc2 is an enzyme that produces tyramine (the precursor of octopamine) from tyrosine, and Tbh is an enzyme involved in octopamine production from tyramine [24,25]. Thus, these two enzymes are required for octopamine production [26], whereas ChAT is an acetylcholine (Ach)-synthesizing enzyme [27]. Furthermore, *Tdc2* and *Tbh* are detected in Pdf neurons as well as LNds [23]. In this study, we identified that octopamine in LNds modulates LTM maintenance, whereas octopamine in Pdf neurons modulates LTM consolidation. These findings indicate that octopamine signaling in distinct clock neuron clusters differentially regulates separate phases of memory processing in *Drosophila*.

## Results

### LNd-specific knockdown of *Tdc2* and *Tbh* induces LTM impairment

In *Drosophila*, LNds consist of six clock neurons per hemisphere [11]. We previously identified two GAL4 lines—*R18H11* and *R78G02*—in which GAL4 is expressed in subsets of these neurons [9]. *R18H11* drives GAL4 expression in two LNds per hemisphere, whereas *R78G02* targets two to three [9]. However, in both GAL4 lines, GAL4 also labels brain neurons other than LNds. To achieve more specific labeling, we generated an LNd-specific split-GAL4 line (hereafter referred to as LNd-split-GAL4) using a combination of the *R18H11-p65.AD* and *R78G02-GAL4.DBD* lines. In LNd-split-GAL4, split GAL4 is expressed in only two LNds per hemisphere (Fig 1C) [9]. Using this line and UAS-*shi*^ts1, we previously confirmed

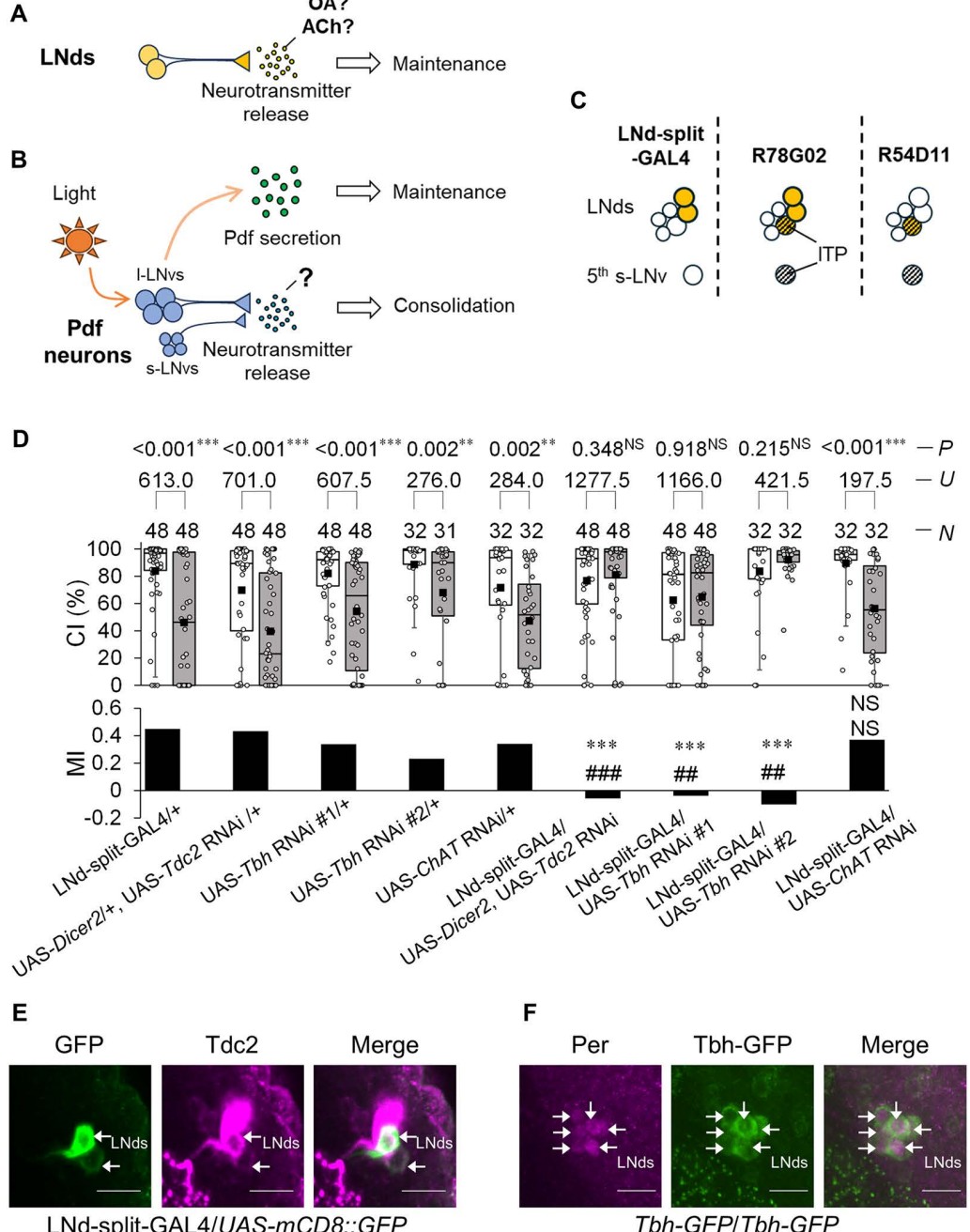

**Fig 1. LNd-specific knockdown in *Tdc2* and *Tbh* induces LTM impairments. (A)** Schematic of LNds-mediated LTM maintenance. **(B)** Schematic of regulation of LTM processes in a Pdf-dependent or Pdf-independent manner. **(C)** Schematic of GAL4-expressing clock neurons in LNd-split-GAL4, *R78G02*, and *R54D11*. **(D)** 5 d memory after 7 h conditioning. LNd-split-GAL4 and four UAS-RNAi lines (UAS-*Dicer2*; UAS-*Tdc2* RNAi, UAS-*Tbh* RNAi #1, UAS-*Tbh* RNAi #2, and UAS-*ChAT* RNAi) were used in the experiments. We visualized the data using a box plot with individual data points (white circles). Box plots for a set of CI data show 5th, 25th, 75th, and 95th centiles. The black square in each box indicates the mean, the line in each box is drawn at the median, white boxes indicate naive males, and gray boxes indicate conditioned males. The asterisks in the figure show the results of comparison with GAL4 control, and the sharp symbols show the results of comparison with UAS control. **, $P < 0.01$; ***, $P < 0.001$; ##, $P < 0.01$; ###, $P < 0.001$; NS, not significant; $P$, probability; $U$, Mann–Whitney $U$; $N$, sample size in each box. **(E)** Confocal section images at LNd level of the adult brain. LNd-split-GAL4/UAS-*mCD8::GFP* males were used. Arrows, LNds. Scale bars represent 10 µm. Magenta, Tdc2; green, mCD8::GFP. **(F)** Confocal section images at the LNd level of the adult brain. Males homozygous for *Tbh-GFP* were used. Arrows, LNds. Scale bars represent 10 µm. Magenta, Period; green, Tbh-GFP.

that the disruption of neurotransmission in LNd-split-GAL4-positive neurons during the LTM maintenance phase impairs LTM [9]. To investigate whether octopamine or acetylcholine (ACh) in LNds contributes to LTM, we knocked down genes involved in their synthesis (*Tdc2*, *Tbh*, and *ChAT*) using LNd-split-GAL4 in combination with multiple UAS-RNAi lines. The efficacy of *Tdc2* RNAi and *Tbh* RNAi was assessed by qRT-PCR analysis using a pan-neuronal driver, *nSyb*-GAL4. *nSyb*-GAL4/UAS-*Tdc2* RNAi males did not show a reduction in *Tdc2* expression levels (S1A Fig). However, the co-expression of UAS-*Dicer2* achieved ~90% knockdown relative to GAL4 control flies (S1B Fig). *nSyb*-GAL4/UAS-*Tbh* RNAi #1 and *nSyb*-GAL4/UAS-*Tbh* RNAi #2 flies showed about a 70% reduction in *Tbh* expression levels compared with GAL4 control flies (S1C, S1D Fig). Unlike *Tbh* and *Tdc2* knockdown flies, *nSyb*-GAL4/UAS-*ChAT* RNAi flies were lethal, precluding qRT-PCR analysis. This lethality is consistent with the finding that *ChAT* mutant flies are also lethal (https://flybase.org/reports/FBgn0000303.htm), suggesting that *ChAT* knockdown using UAS-*ChAT* RNAi reduces cholinergic function.

In LNd-split-GAL4 and UAS control males, the courtship index (CI), an indicator of male courtship activity, of conditioned males was significantly lower than that of naive males on day 5 after 7 h conditioning (Fig 1D). Thus, in these control flies, we detected courtship memory on day 5 after 7 h conditioning (hereafter referred to as '5 d memory'). However, in the $F_1$ hybrid between LNd-split-GAL4 and either UAS-*Dicer2*; UAS-*Tdc2* RNAi, UAS-*Tbh* RNAi #1, or UAS-*Tbh* RNAi #2, no significant differences in CI were detected between naive and conditioned males (Fig 1D). To quantify courtship memory, the memory index (MI) was calculated as described in the Methods section. In these $F_1$ hybrid males, MI was significantly lower than those in LNd-split-GAL4 and UAS control males, indicating that the LNd-specific knockdown of *Tdc2* and *Tbh* impaired LTM (Fig 1D). Tdc2 encodes a tyrosine decarboxylase that converts tyrosine into tyramine, a precursor of octopamine, whereas Tbh catalyzes the conversion of tyramine into octopamine. Thus, these results indicate that reduced octopamine production in two pairs of LNds impairs LTM. However, *Tbh* knockdown in LNds did not affect courtship STM formation (S2A Fig). Unlike *Tdc2* and *Tbh* knockdown in LNds, *ChAT* knockdown in LNds had little effect on 5 d memory after 7 h conditioning (Fig 1D).

As shown in Fig 1D, naive CI in LNd-split-GAL4/UAS-*Tbh* RNAi #1 (mean naive CI = 62%) was significantly lower than that in LNd-split-GAL4 control males (Steel–Dwass test, $q = 3.338$, $P < 0.020$), despite the absence of a significant difference between LNd-split-GAL4/+ and UAS-Tbh RNAi #1/+ males (Steel–Dwass test, $q = 1.60$, $P = 0.795$). We previously confirmed that the wild-type strains with a mean naive CI of less than 40% exhibit courtship suppression on day 5 after 7 h conditioning [6]. Thus, it is unlikely that the difficulty in detecting courtship memory in LNd-split-GAL4/UAS-*Tbh* RNAi #1 males is solely due to a reduction in naive CI. Unlike LNd-split-GAL4/UAS-*Tbh* RNAi #1 males, no significant difference was detected between LNd-split-GAL4 control and LNd-split-GAL4/UAS-*Tbh* RNAi #2 males (Steel–Dwass test, $q = 1.31$, $P = 0.926$). These results suggest that the reduced courtship activity in LNd-split-GAL4/UAS-*Tbh* RNAi #1 naive males may not be due to the *Tbh* knockdown itself, but rather by genetic background effects.

## LNd-specific knockdown of *Tbh* has little effect on sleep and circadian rhythms

In *Drosophila*, sleep is required for the consolidation of courtship LTM [28,29]. To confirm whether LTM impairments induced by LNd-specific *Tbh* knockdown result from altered sleep, we measured the sleep amount in light:Dark (LD) cycles using LNd-split-GAL4/UAS-*Tbh* RNAi #1 males. LNd-split-GAL4/+ and UAS-*Tbh* RNAi #1/+ males were used as the control. For the total sleep amount during the day and night, no significant differences were detected between LNd-split-GAL4/UAS-*Tbh* RNAi #1 males and UAS-*Tbh* RNAi #1/+ control males (S3A-S3C Fig), although significant differences were detected between LNd-split-GAL4/UAS-*Tbh* RNAi #1 males and LNd-split-GAL4/+ control males (S3A-S3C Fig). We also assessed two additional sleep parameters: sleep bout number and sleep bout duration. The sleep bout number during the day in *Tbh* knockdown males was higher than that in GAL4 control males and lower than that in UAS control males (S3D Fig). There was no significant difference in sleep bout number during the night between the GAL4 control males and *Tbh* knockdown males (S3D Fig). The sleep bout duration during the day in *Tbh* knockdown males was shorter than that in GAL4 control males and longer than that in UAS control males (S3E Fig). The sleep bout duration

during the night did not differ significantly between the *Tbh* knockdown males and the GAL4 control males. However, the waking activity index was elevated in *Tbh* knockdown males only during the day (S3F Fig). Thus, *Tbh* knockdown has little effect on the amount and quality of sleep, but it appears to selectively increase spontaneous activity during daytime wakefulness.

We next examined whether LNd-specific *Tbh* knockdown affects circadian behavioral rhythms under constant darkness. No significant differences were detected between LNd-split-GAL4/UAS-*Tbh* RNAi #1 and control males in the proportion of rhythmic individuals (% rhythmic) or circadian periods (S4 Fig). Taken together, it is unlikely that the memory impairment observed in *Tbh* knockdown flies results from alterations in circadian rhythms.

**Temporal knockdown of *Tbh* in *R78G02*-positive cells impairs consolidation and maintenance of LTM**

Are Tdc2 and Tbh actually expressed in LNds? Previous chronobiological studies have shown that Tdc2 is present in a subset of LNds, as revealed by immunohistochemical staining with an anti-Tdc2 antibody [30]. To determine whether Tdc2 is expressed in the specific subset labeled by LNd-split-GAL4, we performed immunohistochemical staining with an anti-Tdc2 antibody, which showed Tdc2 signals in two LNd-split-GAL4-positive neurons per hemisphere (Fig 1E). Unlike Tdc2, antibodies for Tbh were unavailable. To examine Tbh expression in LNds, we utilized a fly-TransgeneOme (fTRG) line (VDRC ID: 318242), which expresses a C-terminal superfolder GFP (sGFP)-tagged fusion protein under the control of the endogenous *Tbh* promoter [31]. In this study, we used a fTRG 318242 line (hereafter referred to as '*Tbh-GFP*') to visualize *Tbh*-expressing neurons. Since the circadian clock gene *period* is expressed in all six LNds [32], we observed all LNds in males homozygous for *Tbh-GFP* using an anti-Per antibody. The co-expression of Per and Tbh-GFP was detected in all LNds (Fig 1F). Therefore, Tbh is likely expressed in LNd-split-GAL4-positive cells.

To investigate whether octopamine production in LNds is required for the consolidation or maintenance of LTM, we performed temporal *Tbh* knockdown experiments using *tub*-GAL80$^{ts}$ [33]. The *tub*-GAL80$^{ts}$ transgene encodes a ubiquitously expressed GAL4 repressor that functions at the permissive temperature (PT, 25 °C) but is inactive at the restrictive temperature (RT, 30 °C). However, since GAL80$^{ts}$ has no effect on split-GAL4-driven gene expression, we used *tub*-GAL80$^{ts}$/UAS-*Tbh* RNAi #1; *R78G02*/+ males in this experiment. *tub*-GAL80$^{ts}$/+; *R78G02*-GAL4/+ and UAS-*Tbh* RNAi #1/+ males were used as the control. *R78G02* expresses GAL4 in three LNds and the 5th s-LNv in each hemisphere [34]. In *Drosophila*, the ion transport peptide (ITP) is expressed in one LNd and the 5th s-LNv per brain hemisphere [35]. To confirm this expression pattern in *R78G02*, we performed immunostaining with an anti-ITP antibody. In *R78G02*/UAS-*mCD8::GFP* males, the one LNd and the 5th s-LNv were ITP-positive in each hemisphere (Fig 2A). We next examined whether Tdc2 is expressed in *R78G02*-positive neurons using an anti-Tdc2 antibody. Tdc2 signals were detected in three LNds and the 5th s-LNv in *R78G02*-positive cells in each hemibrain (Fig 2B and 2C). We further confirmed whether Tbh is expressed in GAL4-expressing cells in *R78G02* using Tbh-GFP. Tbh-GFP signals were detected in LNds and the 5th s-LNv in *R78G02*-positive cells in each hemibrain (S5A and S5B Fig). Furthermore, they were detected in at least two pairs of *R78G02*-positive non-clock neurons located in the posterior–dorsal region (S5C Fig).

To knock down *Tbh* in *R78G02*-positive cells during the memory consolidation, maintenance, or recall phase, we used *tub*-GAL80$^{ts}$/UAS-*Tbh* RNAi #1; *R78G02*-GAL4/+ males. *tub*-GAL80$^{ts}$/+; *R78G02*-GAL4/+ and UAS-*Tbh* RNAi #1/+ males were used as the control. For the temporal knockdown of *Tbh*, the temperature was increased to RT (30 °C) for 24 h during three different experimental periods: starting at 24 h before the end of conditioning (learning/consolidation phase), 48–72 h after conditioning initiation (maintenance phase), and 24 h before the test initiation (recall phase). In all genotypes, LTM was intact at PT (25 °C) (Fig 2D). On the other hand, the temporal knockdown of *Tbh* during the learning/consolidation phase or the maintenance phase significantly impaired LTM (Fig 2E and 2F), whereas that during the recall phase did not (Fig 2G). These findings suggest that octopamine signaling from *R78G02*-positive Tbh-expressing neurons—including a subset of LNds, the 5th s-LNv, and non-clock neurons—plays a critical role in the consolidation and maintenance, but not recall, of courtship LTM.

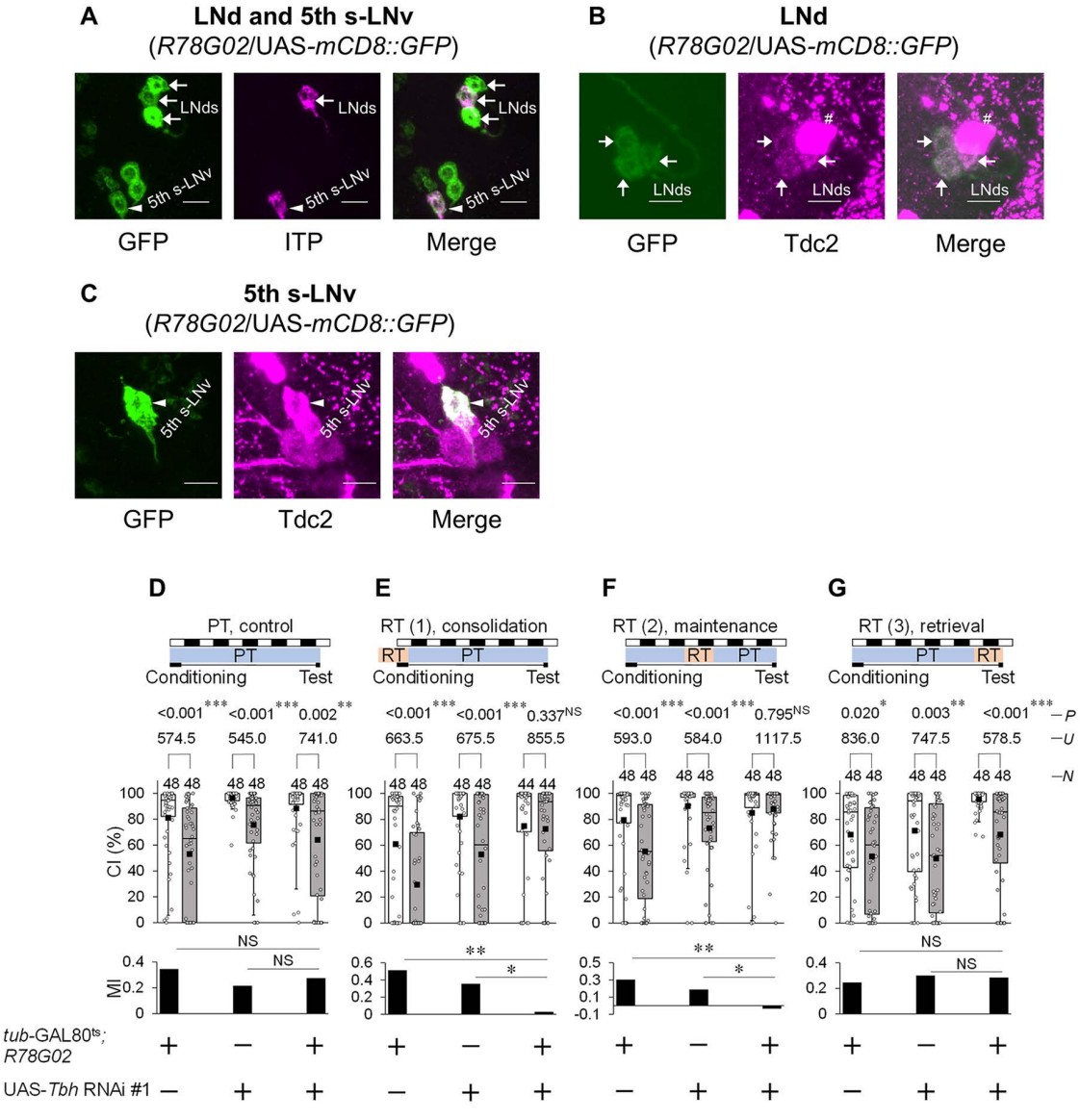

**Fig 2. Temporary *Tbh* knockdown in *R78G02*-positive cells impairs LTM maintenance. (A)** Confocal section images at LNd and 5th s-LNv levels of the adult brain. R78G02/UAS-*mCD8::GFP* males were used. Arrows, LNds; arrow heads, 5th s-LNv. Scale bars represent 10 μm. Magenta, ITP; green, mCD8::GFP. **(B)** Confocal section images at LNd level of the adult brain. R78G02/UAS-*mCD8::GFP* males were used. Scale bars represent 10 μm. Arrows, LNds; #, non-clock neurons; magenta, Tdc2; green, mCD8::GFP. **(C)** Confocal section images at 5th s-LNv level of the adult brain. R78G02/UAS-*mCD8::GFP* males were used. Arrows, LNds. Scale bars represent 10 μm. Magenta, Tdc2; green, mCD8::GFP. **(D–G)** 5 d memory in *tub*-GAL80^ts/UAS-*Tbh* RNAi #1; *R78G02*/+ and control flies. We visualized the data using a box plot with individual data points (white circles). See Fig 1 for an explanation of box plots. *, P< 0.05; **, P< 0.01; ***, P< 0.001; NS, not significant; P, probability; U, Mann–Whitney U; N, sample size in each box. **(D)** All experiments were carried out at PT (25 °C). **(E)** Flies were kept at RT (30 °C) for 24 h before the end of 7 h conditioning. **(F)** Flies were kept at RT for 39 to 63 h after 7 h conditioning. **(G)** Flies were kept at RT for 24 h before the end of test.

## Disruption of neurotransmission in *R78G02*-positive cells impairs the consolidation and maintenance of LTM

To determine whether neurotransmitter release from *R78G02*-positive cells is required for LTM consolidation and maintenance, we experimented with the temporal disruption of neurotransmission using *R78G02*/UAS-*shi^{ts1}* males. *R78G02*/

UAS-*shi*^ts1^ males showed LTM at PT (25 °C) as was observed in GAL4 and UAS control males (Fig 3A). When the temperature was raised to RT (30 °C) during 7 h conditioning, LTM was impaired in *R78G02*/UAS-*shi*^ts1^ males (Fig 3B). In addition, when males were kept at 30 °C for 39–63 h after the end of conditioning, LTM was also impaired (Fig 3C). However, the disruption of synaptic transmission from LNds during the test had little effect on LTM (Fig 3D). Consistent with the experimental results of *Tbh* knockdown, these results indicate that disruption of neurotransmission from *R78G02*-positive cells impairs LTM consolidation and maintenance. Thus, these results also support our idea that octopamine release from *R78G02*-positive cells modulates both LTM consolidation and maintenance.

## Octopamine synthesis in ITP-positive clock neurons has little effect on LTM

We previously found that the disruption of neurotransmission from LNd-split-GAL4-positive cells impairs only LTM maintenance without affecting LTM consolidation [9]. On the other hand, *Tbh* knockdown or blocking neurotransmission in *R78G02*-positive cells impairs both LTM consolidation and maintenance. Notably, LNd-split-GAL4 is expressed in only two LNds in each hemibrain, whereas *R78G02*-positive LNds include an ITP-positive cell in each hemibrain (Fig 2A). We confirmed that LNd-split-GAL4-positive cells do not express ITP (Fig 4A). Moreover, *R78G02*-positive cells include a pair of ITP-positive 5th s-LNvs (Fig 2A). The main differences between GAL4-expressing clock neurons in LNd-split-GAL4 and *R78G02* can be summarized in Fig 1C. Since ITP-positive LNds and 5th s-LNv are present in *R78G02* but absent in LNd-split-GAL4 (Fig 1C), we hypothesized that octopamine in these ITP-positive clock neurons might contribute to LTM consolidation. To examine this possibility, we used a GAL4 line, *R54D11*. In *R54D11*, GAL4 is mainly expressed in one LNd and the 5th s-LNv (Fig 1C) [36]. We confirmed that ITP is expressed in *R54D11*-positive LNds and the 5th s-LNv (Fig 4B). However, the knockdown of either *Tbh* or *Tdc2* in *R54D11*-positive cells did not affect LTM (Fig 4C and 4D). Thus, it seems unlikely that octopamine release from ITP-positive clock neurons affects LTM.

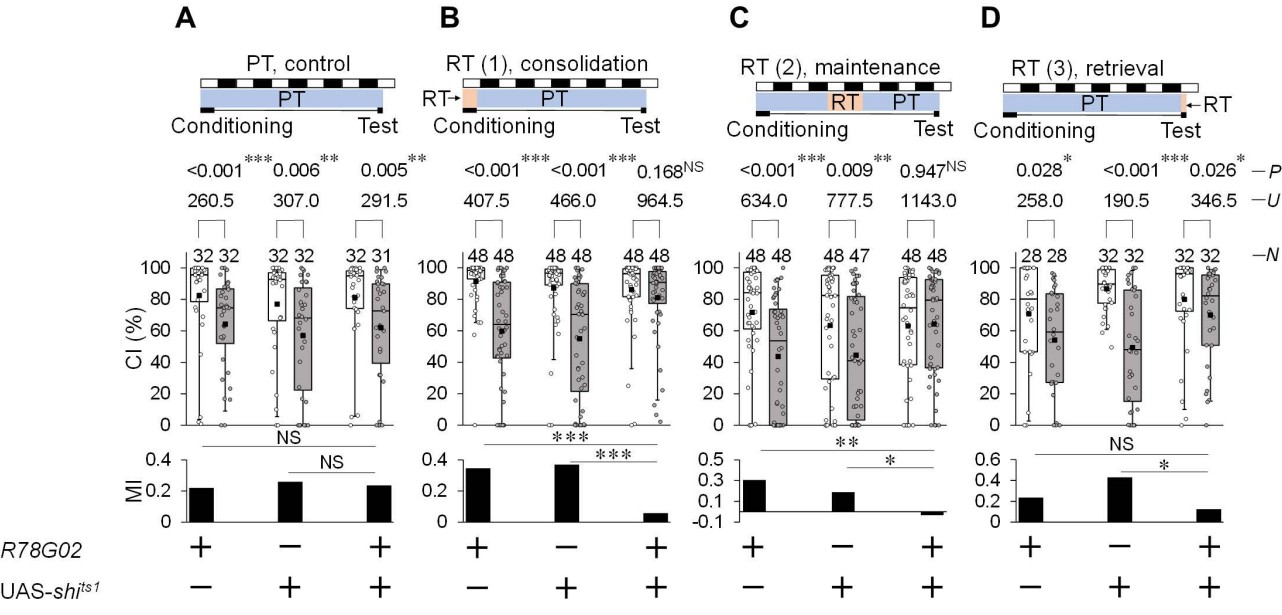

**Fig 3. Disruption of the neurotransmission in GAL4-expressing cells in *R78G02* impairs LTM consolidation and maintenance. (A–D)** 5 d memory in *R78G02*/UAS-*shi*^ts1^ and control flies. We visualized the data using a box plot with individual data points (white circles). See Fig 1 for an explanation of box plots. *, *P* < 0.05; **, *P* < 0.01; ***, *P* < 0.001; NS, not significant; *P*, probability; *U*, Mann–Whitney *U*; *N*, sample size in each box. **(A)** All experiments were carried out at PT (25 °C). **(B)** Flies were kept at RT (30 °C) during 7 h conditioning. **(C)** Flies were kept at RT for 39 to 63 h after 7 h conditioning. **(D)** Flies were kept at RT for 30 min before the test began and until the test ended.

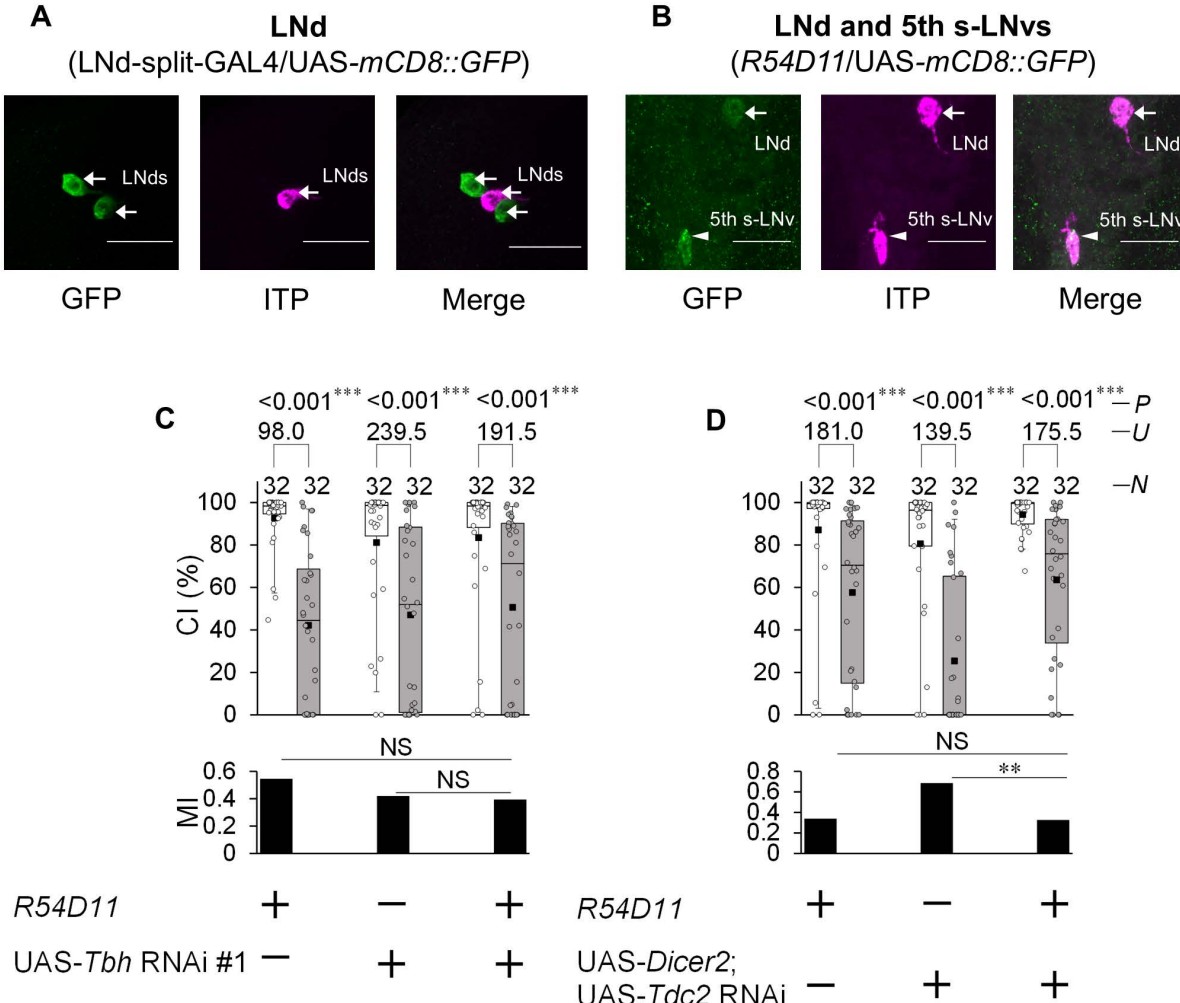

**Fig 4. Octopamine in ITP-positive LNds and 5th s-LNv has little effect on LTM. (A)** Confocal section images at LNd level of the adult brain. Arrows, LNds. Scale bars represent 20 μm. LNd-split-GAL4/UAS-*mCD8::GFP* flies were used. Magenta, ITP; green, mCD8::GFP. **(B)** Confocal section images at LNd and 5th s-LNv level of the adult brain. Arrows, LNds; Arrow heads, 5th s-LNv. Scale bars represent 20 μm. Magenta, ITP; green, mCD8::GFP. **(C)** 5 d memory in *R54D11*/UAS-*Tbh* RNAi #1 and control males. **(D)** 5 d memory in UAS-*Dicer2*/ + ; *R54D11*/UAS-*Tdc2* RNAi and control males. **(C, D)** We visualized the data using a box plot with individual data points (white circles). See Fig 1 for an explanation of box plots. ***, P<0.001; NS, not significant; P, probability; U, Mann–Whitney U; N, sample size in each box.

In summary, octopamine synthesis in the two LNds, excluding ITP-positive LNds, among *R78G02*-positive cells appears to be necessary for LTM maintenance. On the other hand, octopamine synthesis in *R78G02*-positive non-clock neurons may be involved in LTM consolidation.

## Temporal knockdown of *Tbh* in Pdf neurons impairs LTM consolidation

We previously identified that neurotransmission from Pdf neurons is required for LTM consolidation [5]. Tdc2 is also expressed in Pdf neurons [30]. Thus, we investigated whether Tdc2 and Tbh in Pdf neurons are involved in LTM consolidation. First, we confirmed whether Tdc2 is expressed in Pdf-positive l-LNvs and s-LNvs using an anti-Tdc2 antibody (Fig 5A and 5B). In the case of l-LNvs, all 20 brains examined showed Tdc2 signals in four l-LNvs per hemisphere (Fig 5A). In contrast, among the 18 brains analyzed for s-LNvs, although Tdc2 signals were detected in only a single s-LNv in

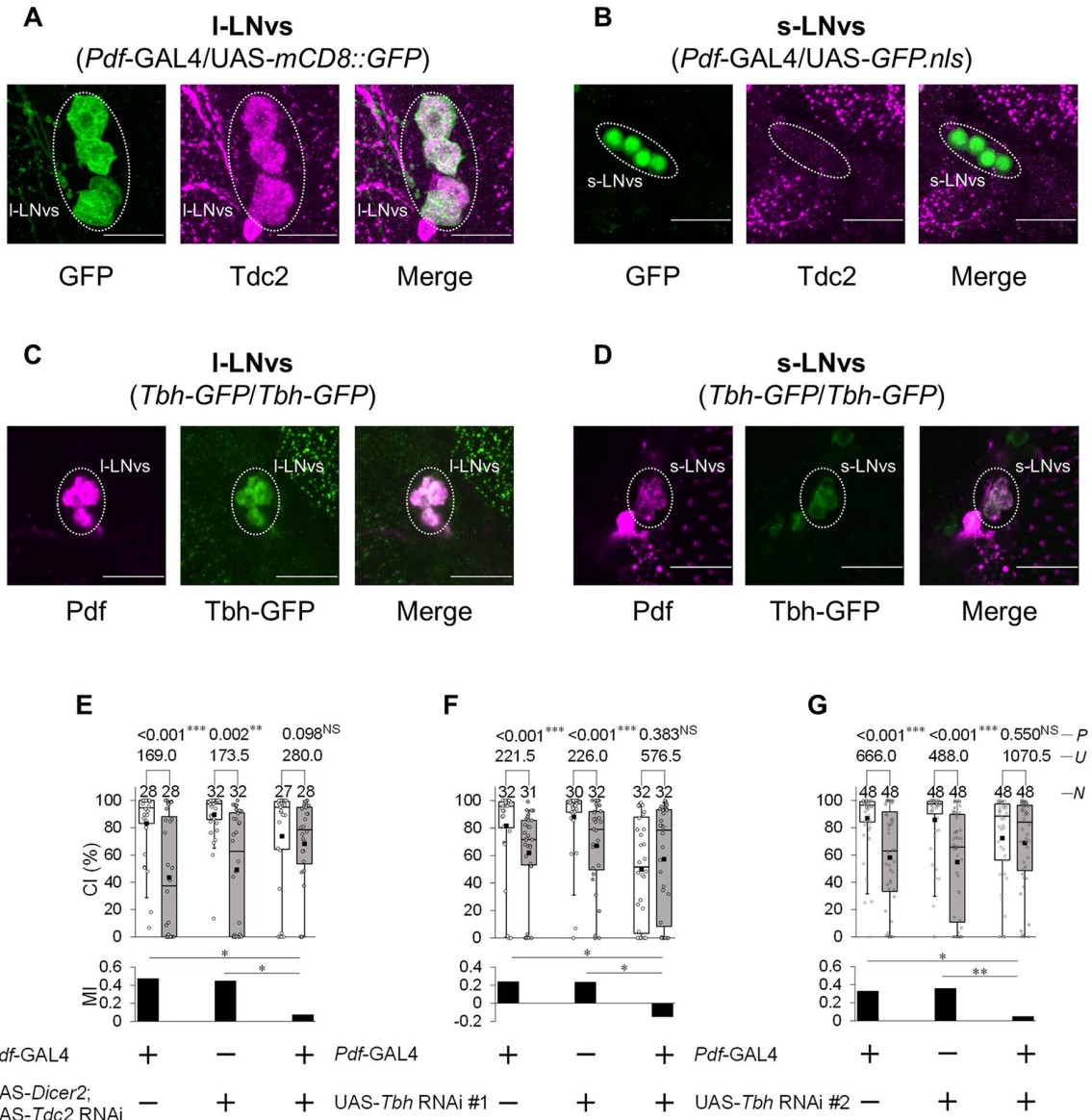

**Fig 5. Inhibition of octopamine synthesis in Pdf neurons induces LTM impairments. (A)** Confocal section images at l-LNv level of the adult brain. Dotted area, l-LNvs. Scale bars represent 20 μm. *Pdf*-GAL4/UAS-*mCD8::GFP* flies were used. Magenta, Tdc2; green, mCD8::GFP. **(B)** Confocal section images at s-LNv level of the adult brain. Dotted area, s-LNvs. Scale bars represent 20 μm. *Pdf*-GAL4/UAS-*GFP.nls* flies were used. Magenta, Tdc2; green, GFP.NLS. **(C)** Confocal section images at l-LNv level of the adult brain. Dotted area, l-LNvs. Scale bars represent 20 μm. Males homozygous for *Tbh-GFP* were used. Magenta, Pdf; green, Tbh-GFP. **(D)** Confocal section images at s-LNv level of the adult brain. Dotted area, s-LNvs. Scale bars represent 20 μm. Males homozygous for *Tbh-GFP* were used. Magenta, Pdf; green, Tbh-GFP. **(E)** 5 d memory in UAS-*Dicer2*/*Pdf*-GAL4; UAS-*Tdc2* RNAi/+ males. Pdf-GAL4/+ and UAS-*Dicer2*/+; UAS-*Tdc2* RNAi/+ males were used as the control. **(F)** 5 d memory in *Pdf*-GAL4/UAS-*Tbh* RNAi #1 males. *Pdf*-GAL4/+ and UAS-*Tbh* RNAi #1/+ males were used as the control. **(E, F)** We visualized the data using a box plot with individual data points (white circles). See Fig 1 for an explanation of box plots. **, *P* < 0.01; ***, *P* < 0.001; NS, not significant; *P*, probability; *U*, Mann–Whitney *U*; *N*, sample size in each box.

one hemisphere of one brain, they were absent in the s-LNvs of nearly all other brains (Fig 5B). Furthermore, to confirm whether Tbh is also expressed in Pdf neurons, Pdf-positive l-LNvs and s-LNvs in *Tbh-GFP* males were visualized using an anti-Pdf antibody. In *Tbh-GFP* males, strong GFP signals were detected in all l-LNvs, whereas weak GFP signals were

detected in s-LNvs (Fig 5C and 5D). Taken together, these results suggest that octopamine is synthesized in Pdf neurons, and that synthesis in l-LNvs among Pdf neurons is particularly prominent.

We next investigated the effects of *Tdc2* and *Tbh* knockdown on LTM. In males with knockdown of *Tdc2* or *Tbh* in Pdf neurons, MI was significantly reduced compared with those in GAL4 and UAS control males (Fig 5E–5G). In Fig 5F and 5G, the naïve CI in *Pdf*-GAL4/UAS-*Tbh* RNAi #1 and *Pdf*-GAL4/UAS-*Tbh* RNAi #2 males was significantly lower than that in GAL or UAS control (*Pdf*-GAL4/+ vs *Pdf*-GAL4/UAS-*Tbh* RNAi #1, Steel–Dwass test, $q = 3.90$, $P < 0.001$; UAS-*Tbh* RNAi #1/+ vs *Pdf*-GAL4/UAS-*Tbh* RNAi #1, Steel–Dwass test, $q = 4.77$, $P < 0.001$; *Pdf*-GAL4/+ vs *Pdf*-GAL4/UAS-*Tbh* RNAi #2, Steel–Dwass test, $q = 2.60$, $P < 0.05$; UAS-*Tbh* RNAi #2/+ vs *Pdf*-GAL4/UAS-*Tbh* RNAi #2, Steel–Dwass test, $q = 3.14$, $P < 0.01$). These findings suggest that octopamine in Pdf neurons plays a direct role in regulating male courtship activity. However, *Tbh* knockdown in Pdf neurons did not affect courtship STM (S2B Fig), suggesting that the reduction in CI by *Tbh* knockdown does not interfere with courtship STM formation.

To test whether octopamine production in Pdf neurons is required for LTM consolidation, we performed temporal knockdown experiments using *Pdf*-GAL4/UAS-*Tbh* RNAi #1; *tub*-GAL80$^{ts}$/+ males. *Pdf*-GAL4/+; *tub*-GAL80$^{ts}$/+ and UAS-*Tbh* RNAi #1/+ males were used as the control. In the experiments, the PT was 22°C outside the test phase and 25°C during the test phase. In all genotypes, LTM was intact at PT on day 5 after 7 h conditioning (Fig 6A). When males were kept at RT (30 °C) during the period from 17 h before the initiation of conditioning to the end of conditioning, LTM was impaired (Fig 6B). In addition, no significant differences in the CI of naive males were detected between GAL4 control and *Tbh* knockdown males (*Pdf*-GAL4/+; *tub*-GAL80$^{ts}$/+ vs *Pdf*-GAL4/UAS-*Tbh* RNAi #1; *tub*-GAL80$^{ts}$/+; Steel–Dwass test, $q = 0.878$, $P = 0.654$; UAS-*Tbh* RNAi #1/+ vs *Pdf*-GAL4/UAS-*Tbh* RNAi #1; *tub*-GAL80$^{ts}$/+, Steel–Dwass test, q = 2.906,

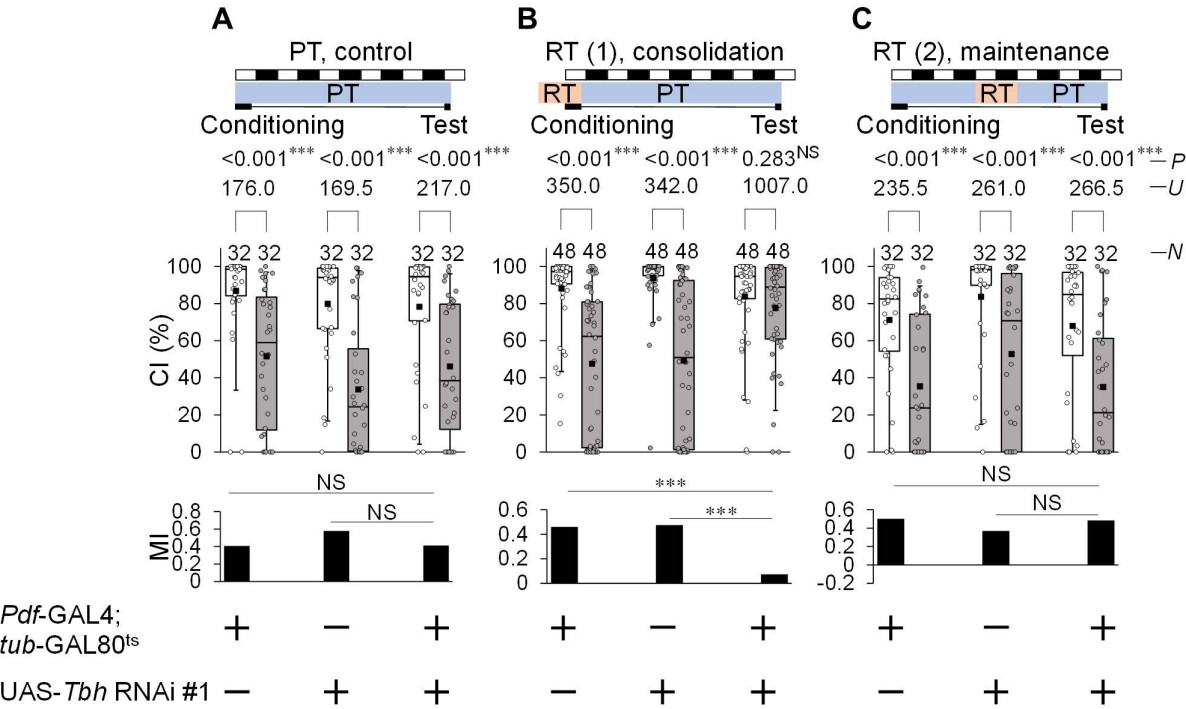

**Fig 6. Temporary *Tbh* knockdown in Pdf neurons impairs LTM consolidation. (A)** All experiments were carried out at PT (22–25 °C). **(B)** Flies were kept at RT (30 °C) for 24 h before the end of 7 h conditioning. **(C)** Flies were kept at RT for 39 to 63 h after 7 h conditioning. **(A–C)** 5 d memory in *Pdf*-GAL4/UAS-*Tbh* RNAi #1; *tub*-GAL80$^{ts}$/+ and control males. We visualized the data using a box plot with individual data points (white circles). See Fig 1 for an explanation of box plots. ***, $P < 0.001$; NS, not significant; P, probability; U, Mann–Whitney U; N, sample size in each box.

*P* < 0.05). However, raising the temperature to 30 °C during the maintenance phase (39–63 h after conditioning) had little effect on LTM (Fig 6C) and the CI of naive males (*Pdf*-GAL4/ + ; *tub*-GAL80^ts^/ + vs *Pdf*-GAL4/UAS-*Tbh* RNAi #1; *tub*-GAL80^ts^/ + ; Steel–Dwass test, q = 0.175, *P* = 0.983; UAS-*Tbh* RNAi #1/ + vs *Pdf*-GAL4/UAS-*Tbh* RNAi #1; *tub*-GAL80^ts^/ + , Steel–Dwass test, q = 2.526, *P* < 0.05). Since we previously confirmed that neurotransmission from Pdf neurons does not affect LTM recall [5], we did not perform recall-phase-specific *Tbh* knockdown. Taken together with previous findings, these results suggest that octopamine signaling in Pdf neurons mainly contributes to LTM consolidation but not LTM maintenance or recall.

The CI of naive males in *Pdf*-GAL4/UAS-*Tbh* RNAi #1 was significantly lower than that in GAL or UAS control males (Fig 5). However, in the temporal knockdown experiments, no significant difference was detected between the CI of the *Tbh* knockdown males and that of the GAL4 control males. (Fig 6). Thus, the reduced courtship activity observed in *Pdf*-GAL4/UAS-*Tbh* RNAi #1 males is likely due to developmental defects caused by the suppression of *Tbh* expression.

### *Tbh* knockdown in Pdf neurons has little effect on sleep and circadian rhythms

Pdf-expressing l-LNvs are essential for light-dependent arousal in *Drosophila* [16]. Thus, we examined whether *Tbh* knockdown in Pdf neurons affects the sleep phenotype using *Pdf*-GAL4/UAS-*Tbh* RNAi #1 males. *Pdf*-GAL4/+ and UAS-*Tbh* RNAi #1/ + males were used as the control. Regarding the total sleep amount during the day, no significant differences were detected between *Pdf*-GAL4/UAS-*Tbh* RNAi #1 males and UAS-*Tbh* RNAi #1/ + control males (S6A and S6B Fig), whereas significant differences were detected between *Pdf*-GAL4/UAS-*Tbh* RNAi #1 males and *Pdf*-GAL4/ + control males (S6A and S6B Fig). Regarding the total sleep amount during the night, no significant differences were detected among the three genotypes (S6A and S6C Fig). Regarding both sleep bout number and sleep bout duration, no significant differences were detected between *Tbh* knockdown males and UAS control males (S6D and S6E Fig). Regarding the waking activity index, no significant differences were detected between *Tbh* knockdown males and UAS control males (S6F Fig). Thus, octopamine signaling in Pdf neurons does not appear to have a major impact on the sleep phenotype or spontaneous activity.

Huang et al. reported that *Tdc2* knockdown in Pdf neurons induces arrhythmic locomotor activity [30]. However, it remains unclarified whether Tbh in Pdf neurons is also necessary for rhythmic locomotor activity. To address this possibility, we examined whether *Tbh* knockdown in Pdf neurons affects circadian behavioral rhythms in constant darkness. Unlike *Tdc2* knockdown, *Tbh* knockdown in Pdf neurons did not affect the percentage of rhythmic flies or circadian periods (Fig 7A–7C). Thus, it is unlikely that octopamine in Pdf neurons affects circadian rhythms. However, about 60% of the 55 *Tdc2* knockdown males had arrhythmic locomotor activity (Fig 7D–7G), indicating that Tdc2 in Pdf neurons is required for circadian behavioral rhythms. This finding is consistent with the previous finding [30]. Taken together, tyramine, rather than octopamine, seems to be the key neurotransmitter released from Pdf neurons in regulating circadian rhythms.

## Discussion

Octopamine, a functional analog of vertebrate norepinephrine [24], plays a key role in regulating various behaviors and physiological processes in invertebrates, including feeding, sleep, memory, aggression, and oviposition [26,37–41]. Since Tbh and Tdc2 are broadly expressed in the fly brain [37], octopamine-mediated neurotransmission is considered to be a major modulator of neuronal activity and behavior. In the *Drosophila* brain, Tdc2 is expressed in a subset of clock neurons including Pdf neurons and LNds, and the expression levels of *Tdc2* mRNA and its protein are cycled in a circadian manner [30]. Since the knockdown of *Tdc2* in Pdf neurons induces arrhythmic locomotor activity [30], tyramine or octopamine in Pdf neurons may contribute to the generation of circadian behavioral rhythms. Consistent with the previous report [30], we also confirmed that *Tdc2* knockdown in Pdf neurons induces arrhythmic locomotor activity (Fig 7). Nevertheless, *Tbh* knockdown in Pdf neurons had negligible effects on circadian behavioral rhythms (Fig 7). These findings suggest that tyramine, rather than octopamine, in Pdf neurons is primarily responsible for regulating circadian rhythms. However, the

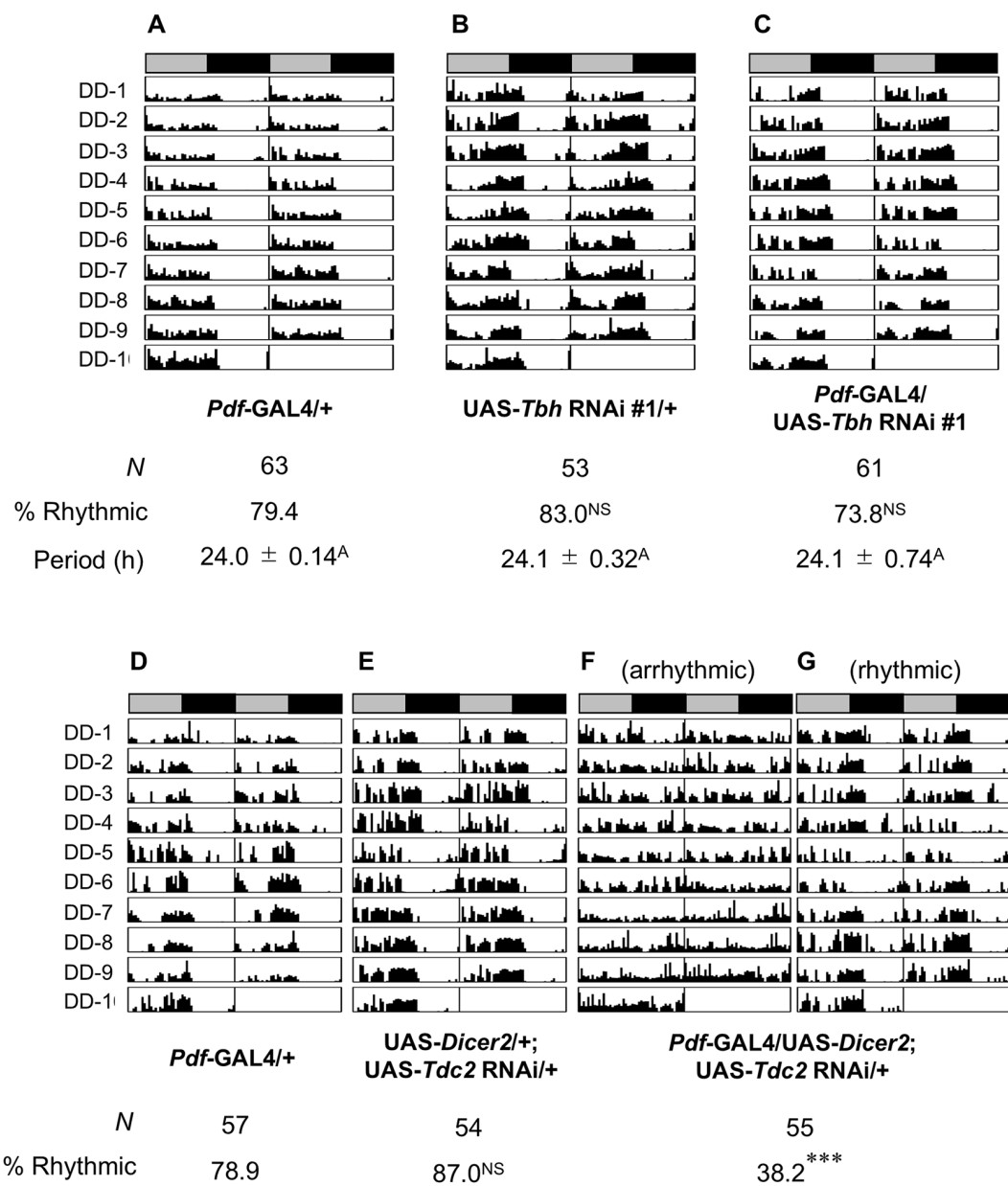

**Fig 7. Expression of *Tdc2*, but not *Tbh*, in Pdf neurons is required for circadian behavioral rhythms. (A–G)** Double-plotted actogram in DD. The bar above each actogram indicates subjective day (gray) and night (black). Each actogram shows locomotor activity for 10 days in DD at 25◦C. We used the Kruskal–Wallis test for comparisons of the circadian period. The same letters in superscripts indicate no significant difference ($P > 0.05$). *N*, sample size. **(A–C)** *Pdf*-GAL4/UAS-*Tbh* RNAi #1 males were used. *Pdf*-GAL4/+ and UAS-*Tbh* RNAi #1/+ males were used as the control. *N*, sample size. For the percentage of rhythmicity (% rhythmic), UAS-*Tbh* RNAi #1/+ or *Pdf*-GAL4/UAS-*Tbh* RNAi #1 males were compared with GAL4 control males. **(D–G)** UAS-*Dicer2*/*Pdf*-GAL4; UAS-*Tdc2* RNAi/+ males were used. *Pdf*-GAL4/+ and UAS-*Dicer2*/+; UAS-*Tdc2* RNAi/+ males were used as the control. *N*, sample size. For the percentage of rhythmicity (% rhythmic), UAS control males or $F_1$ males between the GAL4 and UAS lines were compared with GAL4 control males.

role of octopamine synthesis in clock neurons in regulating brain functions beyond circadian control had not yet been identified. In this study, we found that the LNd-specific knockdown of *Tbh* and *Tdc2* induces LTM impairment (Fig 1), and the Pdf-neuron-specific knockdown of *Tbh* and *Tdc2* also induces LTM impairment (Fig 5). Furthermore, we confirmed that octopamine synthesis in LNds and Pdf neurons does not significantly affect sleep (S3 and S6 Figs). Thus, we demonstrated for the first time that octopamine synthesis in clock neurons is essential for courtship LTM without significantly affecting circadian behavioral rhythms and sleep in *Drosophila*.

Immunostaining with an anti-Tdc2 antibody revealed Tdc2 signals in a subset of LNds (Figs 1E and 2B), whereas Tbh-GFP signals were detected in all six LNds (Fig 1F). Similarly, in Pdf neurons, Tdc2 signals were primarily detected in l-LNvs, with little to no signal detected in the s-LNvs except in a single cell (Fig 5). On the other hand, Tbh-GFP males revealed strong Tbh-GFP signals in all four l-LNvs per hemisphere and weak Tbh-GFP signals in all four s-LNvs per hemisphere (Fig 5). These results suggest that the anti-Tdc2 antibody may have limited sensitivity and might not detect Tdc2 in cells expressing low Tdc2 levels. Nevertheless, in LNd-split-GAL4, both the anti-Tdc2 antibody and Tbh-GFP signals were detected in two pairs of LNds, indicating that both Tdc2 and Tbh are expressed in LNd-split-GAL4-positive LNds.

In LNd-split-GAL4, split-GAL4 expression is restricted to only two pairs of LNds, and the blockade of neurotransmission in these neurons selectively impairs LTM maintenance [9]. Since Tbh is expressed in all LNds (Fig 1), LNd-split-GAL4-positive neurons should be Tbh-positive. Furthermore, the LNd-specific knockdown of *Tbh* or *Tdc2* using LNd-split-GAL4 resulted in LTM impairment (Fig 1). These findings suggest that octopamine-mediated neurotransmission from at least two pairs of LNds plays a critical role in the maintenance of courtship LTM (Fig 8A). In *Drosophila*, MB neurons are considered crucial for courtship LTM, raising the possibility that octopamine release from LNds may modulate MB functions to maintain LTM. However, we were unable to find direct projections from LNds to MB neurons in the database of *Drosophila* connectomics (*neu*Print; https://neuprint.janelia.org/?dataset=hemibrain%3Av1.0.1&qt=findneurons), suggesting that interneurons likely relay information from LNds to the MB. As a next step towards elucidating the molecular and circuit mechanisms underlying LTM maintenance, it will be interesting to identify the octopamine receptors necessary for this process and then to determine which neurons expressing these receptors are required for LTM maintenance.

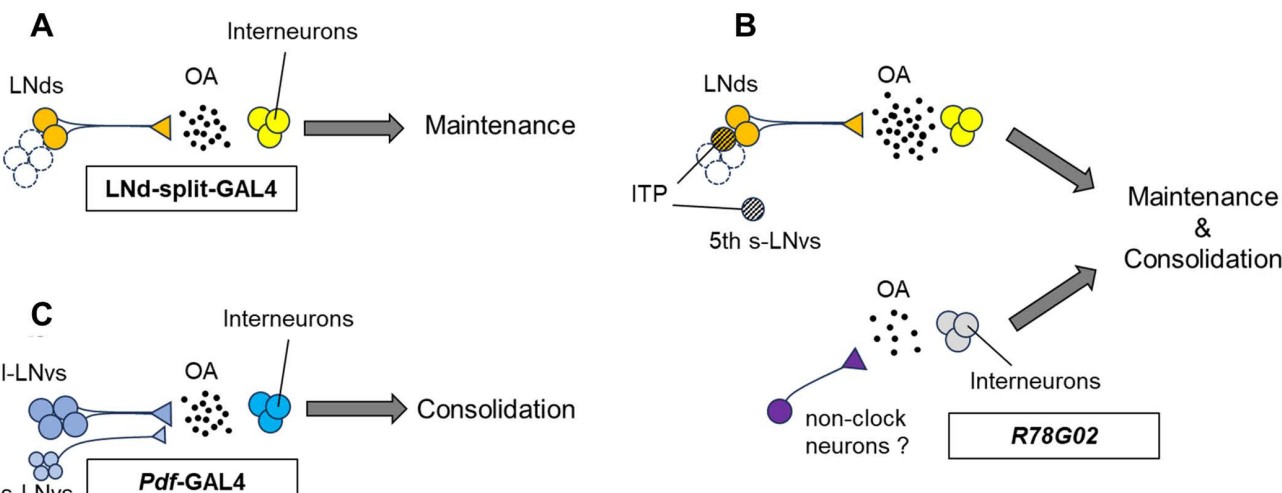

**Fig 8. Possible model of octopamine neurotransmission from clock neurons regulating the consolidation and maintenance of LTM. (A)** Octopamine-mediated neurotransmission from at least two pairs of LNds plays a critical role in the maintenance of courtship LTM. **(B)** Octopamine-mediated neurotransmission from *R78G02*-positive LNds—excluding ITP-positive neurons—contributes to LTM maintenance. In contrast, octopamine signaling from *R78G02*-positive non-clock neurons may regulate LTM consolidation. **(C)** Octopamine signaling from Pdf neurons is essential for LTM consolidation.

In *R78G02*, GAL4 is expressed in two pairs of LNds, one pair of ITP-positive LNds, one pair of 5th s-LNvs, and multiple non-clock neurons (Figs 2 and S5). Among the GAL4-expressing neurons in *R78G02*, three pairs of LNds and one pair of 5th s-LNvs were Tdc2-positive (Fig 2). The temporal knockdown of *Tbh* in *R78G02* resulted in impairments in both LTM consolidation and maintenance (Fig 2). However, *Tbh* knockdown in ITP-positive LNds and the 5th s-LNv using *R54D11* did not affect LTM (Fig 4). Taken together, the impairment of LTM maintenance should be due to the inhibition of octopamine synthesis in at least two pairs of LNds in *R78G02* (Fig 8B). However, the cause of the observed LTM consolidation deficit remains unclear. Since neither LNds nor ITP-positive clock neurons are likely to mediate this effect, it is plausible that octopamine release from non-clock neurons within the *R78G02*-positive neurons is responsible for the observed impairment in LTM consolidation. Identifying which of these neurons contributes to memory consolidation will be important for mapping the underlying circuitry of *Drosophila* courtship LTM. Tbh-GFP signals were detected in at least four neurons located in the dorsoposterior region within the R78G02-positive non-clock neurons (S5C Fig). These non-clock neurons may be candidates for mediating the octopaminergic regulation of LTM consolidation.

We previously reported that Pdf neurons have two distinct roles in *Drosophila* courtship LTM [3]. Pdf secretion from l-LNvs is specifically required for LTM maintenance (Fig 1B) [8], whereas Dynamin-dependent neurotransmission from Pdf neurons is critically involved in LTM consolidation (Fig 1B) [5]. However, the classical neurotransmitter released from the Pdf neurons involved in LTM consolidation had not been identified. In this study, we confirmed Tdc2 signals in Pdf-positive l-LNvs but not s-LNvs and Tbh-GFP signals in s-LNvs and l-LNvs (Fig 5). The knockdown of either *Tdc2* or *Tbh* specifically in Pdf neurons resulted in LTM impairment (Fig 5). Furthermore, the temporal knockdown of *Tbh* in Pdf neurons during the consolidation phase also disrupted LTM (Fig 6), suggesting that octopaminergic neurotransmission from Pdf neurons is essential for LTM consolidation (Fig 8C). As with LNds, there is currently no anatomical evidence for direct synaptic connections between Pdf neurons and MB neurons [3]. Thus, identifying the interneurons expressing the octopamine receptors necessary for information transmission from Pdf neurons to MB neurons may be a breakthrough in elucidating the neural network between Pdf neurons and MB neurons.

We previously demonstrated that environmental light plays a critical role in LTM maintenance by activating the transcriptional factor CREB in MBs via Pdf signaling. In this study, we found that octopamine neurotransmission from a single pair of LNds is essential for LTM maintenance. However, it remains elusive whether octopamine signaling from LNds and Pdf signaling from Pdf neurons function as part of a sequential pathway or operate independently. Among the six LNds, the Pdf receptor (Pdfr) is expressed in three LNds [42,43], and LNd-split-GAL4-expressing neurons are Pdfr-positive [44]. Therefore, elucidating whether light-dependent Pdf release from l-LNvs regulates octopamine release from LNds will be an important step toward understanding the neural circuitry underlying LTM maintenance.

*Drosophila* has four distinct octopamine receptors (Oamb, Octβ1R, Octβ2R, and Octβ3R) [45]. Some of these receptors play important roles in memory processing. Octβ2R is expressed in MB [46] and is involved in the octopamine-dependent reinforcement of appetitive memory [47] as well as anesthesia-resistant aversive memory [46]. Oamb, which is primarily expressed in MB, is involved in aversive and appetitive olfactory memory in *Drosophila* [48,49]. Octopamine signaling is also essential for courtship STM. Zhou et al. reported that deletion mutations of *Tbh* and *Oamb* inhibit immediate memory after 1 h courtship conditioning, and specific Tdc2-positive neurons project to the MB neurons [50]. On the other hand, our findings reveal that Tdc2- and Tbh-positive Pdf neurons and LNds are critically involved in regulating different aspects of courtship LTM. Since it seems unlikely that Pdf neurons and LNds project directly to the MB, neural circuits different from the octopamine signaling pathway identified by Zhou et al. should be involved in the consolidation and maintenance of courtship LTM.

In memory research using *Drosophila* courtship conditioning, many genes essential for LTM have been identified [3]. Many of these genes are expressed in MB neurons, whereas others are found in clock neurons [3]. These findings suggest that both the consolidation and maintenance of courtship LTM are regulated by a complex neural network involving interactions between clock neurons and MB neurons [3]. Furthermore, clock genes have also been implicated in the

consolidation of aversive associative olfactory memory [51–53]. Therefore, the complex neural network composed of clock and MB neurons may be involved in processing various types of memory in *Drosophila*. The findings presented here offer new insights into the neural circuits underlying LTM and will aid future efforts to map the specific pathways responsible for memory processing in the *Drosophila* brain.

## Materials and methods

### Fly stocks

All flies were raised on glucose–yeast–cornmeal medium in 12:12 LD cycles at 25.0 ± 0.5°C (45%– 60% relative humidity). Virgin males and females were collected within 8 h after eclosion. The fly stocks used for this study were as follows: wild-type Canton- S (CS), *R78G02* [40010, Bloomington Drosophila Stock Center (BDSC)], *R18H11*-AD (68852, BDSC), *R78G02*-DBD (69904, BDSC), UAS-*mCD8::GFP* (5137, BDSC), UAS-*shi*$^{ts1}$(a gift from Dr. Kitamoto), UAS-*Tdc2* RNAi (25871, BDSC), UAS-*Tbh* RNAi #1 (67968, BDSC), UAS-*Tbh* RNAi #2 (76062, BDSC), UAS-*ChAT* RNAi (60028, BDSC), *tub*-GAL80$^{ts}$ (7017, 7019, BDSC), UAS-*Dicer2* (24650, BDSC), *R54D11* (41279, BDSC), *nSyb*-GAL4 (51635, BDSC), UAS-*mCherry.NLS* (38425, BDSC), UAS-*GFP.NLS* (24775, BDSC), *Pdf*-GAL4 (6900, BDSC), and *Tbh-GFP* [318242, Vienna Drosophila Resource Center (VDRC)]. All lines except for UAS-*Tdc2* RNAi, UAS-*Tbh* RNAi #1, UAS-*Tbh* RNAi #2, UAS-*ChAT* RNAi, and *R54D11* were outcrossed for at least 6 generations to *white*$^{1118}$ flies with the CS genetic background. *R18H11*-AD and *R78G02*-DBD were used to produce an LNd-split-GAL4 line [9].

In this study, only UAS-*Tdc2* RNAi was used in combination with UAS-*Dicer2*. Unlike UAS-*Tdc2* RNAi, the TRiP RNAi lines generated using the VALIUM20 vector in this study (UAS-*Tbh* RNAi #1, UAS-*Tbh* RNAi #2, and UAS-*ChAT* RNAi) are expected to be effective even without being combined with UAS-*Dicer2* (https://bdsc.indiana.edu/stocks/rnai/rnai_all.html). Thus, they were used without being combined with UAS-*Dicer2*.

### Courtship conditioning assay

The courtship conditioning assay was carried out as previously described [5,9]. For LTM, a virgin male (3–6 d old) was placed with a mated female (4–7 d old) in a conditioning chamber (15 mm diameter × 5 mm depth) containing food for 7 h either with (conditioned) or without (naive) a single premated female (7 h conditioning). After 7 h conditioning, only flies showing courtship behaviors toward the mated female were transferred to a glass tube with food (12 mm diameter × 75 mm depth) and kept in isolation for 5 d until the test. The test was performed using a freeze-killed virgin female in a test chamber (15 mm diameter × 3 mm depth). All procedures in the experiments were performed at 25 ± 1.0°C (45–60% relative humidity) except for the temperature shift experiments. The CI was used to quantify the male courtship behaviors of individual flies and was calculated manually. CI is defined as the percentage of time spent performing courtship behaviors during 10 min. We first measured CI in conditioned and naive males ($CI_{Conditioned}$ and $CI_{Naive}$, respectively), and then the mean $CI_{Naive}$ and mean $CI_{Conditioned}$ were calculated. To quantify courtship memory as previously reported [5,8,9,21], MI was calculated using the following formula: MI = (mean $CI_{Naive}$ - mean $CI_{Conditioned}$)/mean $CI_{Naive}$.

### Temporal disruption of neurotransmission using *shi*ts1

Neurotransmission was temporally disrupted as previously described [5]. Briefly, control males (GAL4/+ and UAS-*shi*$^{ts1}$/+) and *shi*$^{ts1}$-expressing males were used in the experiments. They were kept at PT (25 °C) until the temperature shift experiments. To disrupt neurotransmission in GAL4-expressing neurons during the memory consolidation, maintenance, or recall phase, the temperature was increased to RT (30 °C) during three experimental periods: 7 h conditioning (memory consolidation phase), 39–63 h after the end of conditioning (memory maintenance phase), and 30 min before the test began and until the test ended (recall phase).

 

## Temporal knockdown of gene expression using the TARGET system

Gene expression using the TARGET system [33] was temporally knocked down as previously described [5,8]. The transgene *tub*-GAL80<sup>ts</sup> used in the TARGET system encodes a ubiquitously expressed, temperature-sensitive GAL4 repressor that is active at PT but not at RT. In Fig 2, the PT was 25°C. In Fig 6, the PT was 22°C outside the test phase and 25°C during the test phase. Using UAS-*Tbh* RNAi#1 combined with the TARGET system, we knocked down *Tbh* in GAL4-positive neurons at RT (30°C). In these experiments, we shifted PT to RT and vice versa during three experimental phases: 24 h before the end of conditioning, 39–63 h after conditioning, and 24 h before the end of test.

## Real-time quantitative reverse-transcription PCR (qRT-PCR)

The effectiveness of *Tdc2* RNAi and *Tbh* RNAi was confirmed by qRT-PCR using a pan-neural GAL4 line, *nSyb*-GAL4 as previously described [9]. TRizol (Invitrogen, USA) was used to collect total RNA from about 30 male fly heads in each genotype. cDNA was synthesized by the reverse transcription reaction using a PrimeScript RT reagent kit (RR047A, TAKARA, Japan). qRT-PCR was carried out using a LightCycler 96 (Roche, Switzerland) and the THUNDERBIRD SYBR qPCR Mix (QPS-201, TOYOBO, Japan). The primer sequences used for qRT-PCR were as follows: *Tdc2*-Forward, 5′- GAAG GGTTCCGACAAGCTGAATG- 3′; *Tdc2*-Reverse, 5′- TAGTCAATGTCCTCAGCGGTCG- 3′; *Tbh*-Forward, 5′- ACTA CTATCCGGCCACCAAACTG- 3′; *Tbh*-Reverse, 5′- ATGCTCCGGTAATTGGACGACC- 3′; *rp49*-Forward,

5′- AAGATCGTGAAGAAGCGCAC- 3′; *rp49*- Reverse, 5′- TGTGCACCAGGAACTTCTTG- 3′. The expression level of each mRNA was normalized to that of *rp49* mRNA. The average normalized mRNA expression levels in control flies were calculated using data from five independent assays.

## Anti-ITP antibody

The antibody against ITP was generated in guinea pigs by Scrum Inc. (Tokyo, Japan). A peptide corresponding to the C-terminal part of ITP (EMDKYNEWRDTL) was chemically synthesized and coupled at the N-terminus to keyhole limpet hemocyanin. The synthesized peptide was then used as an antigen to immunize guinea pigs by a conventional method. The specificity of the anti-ITP antibody was validated as previously reported [54].

## Immunohistochemistry

Immunohistochemistry was performed as previously described with some modifications [9]. For Per staining, brains were stained with a rabbit anti-Per antibody (MBS610604, MyBioSource, 1/1000) followed by Alexa Fluor 568 anti-rabbit IgG (A11011, Thermo Fisher Scientific, USA) as the secondary antibody (1:500). For Tdc2 staining, brains were stained with a rabbit anti-Tdc2 antibody (ab128225, Abcam, 1/250 or pAB0822-P, Covalab, 1/250) followed by Alexa Fluor Plus 555 anti-rabbit IgG (A32732, Thermo Fisher Scientific, USA) as the secondary antibody (1:1000). For Pdf staining, brains were stained with a mouse anti-Pdf antibody (PDF C7-s, Developmental Studies Hybridoma Bank at the University of Iowa, 1:200) followed by Alexa Fluor 568 anti-mouse IgG (A11004, Thermo Fisher Scientific) as the secondary antibody (1:1000). For GFP staining, brains were stained with a chicken anti-GFP antibody (ab13970, Abcam, USA; 1:1000), followed by Alexa Fluor 488 anti-chicken IgG (A11039, Thermo Fisher Scientific, USA; 1:1000) as the secondary antibody. For ITP staining, brains were stained with a guinea pig anti-ITP antibody (1:20000), followed by Alexa Fluor 568 anti-guinea pig IgG (A-11075, Thermo Fisher Scientific, USA; 1:1000) or Alexa Fluor 647 anti-guinea pig IgG (A-21450, Thermo Fisher Scientific, USA; 1:1000) as the secondary antibody.

To investigate whether cells co-localizing with Tbh and ITP exist among *R78G02*-positive cells, UAS-*mCherry.NLS*/ + ; *R78G02*/*Tbh-GFP* males were used. Thus, we performed triple staining using an anti-DsRed antibody (632496, Clontech, 1/1000), an anti-GFP antibody (ab13970, Abcam, USA; 1:1000), and an anti-ITP antibody (1:20000). The following antibodies were used as secondary antibodies: Alexa Fluor Plus 555 anti-rabbit IgG (A32732, Thermo Fisher Scientific, USA),

Alexa Fluor 488 anti-chicken IgG (A11039, Thermo Fisher Scientific, USA; 1:1000), and Alexa Fluor 647 anti-guinea pig IgG (A-21450, Thermo Fisher Scientific, USA; 1:1000). Adult male brains were fixed in PBS containing 4% formaldehyde for about 60 min at room temperature. After three washes in PBST (0.2% Triton X-100 in PBS), the brain samples were incubated for 1 h in 1% normal goat serum in PBST for blocking and then incubated with the three primary antibodies for 2 days at 4 °C. Next, they were incubated with the three secondary antibodies for 2 days at 4 °C after three washes in PBST. Fluorescence signals were observed under a confocal microscope (C2 or AX, Nikon, Japan).

### Sleep analysis

Measurements and analysis of sleep were performed as previously described with some modifications [55]. The DAM system (Trikinetics) was used for sleep measurements. Sleep was measured for 3 days in 12-h L:12-h D (LD) cycles (lights on at 8:30) at 25 °C. We used 5–7-d-old males at the start of sleep measurements. Locomotor activity data were collected at 1-min intervals for 3 days and analyzed with a Microsoft Excel-based program as described previously [56]. Total sleep amount, sleep bout number, sleep bout durations, and waking activity index were analyzed for each 12-h period of LD or DD and averaged over 3 days for each condition.

### Circadian rhythms of locomotor activity

Measurements and analysis of circadian behavioral rhythms were performed as previously described with some modifications [22]. The flies were entrained to 12:12 LD cycles during their development, and 8–10-d-old single males were placed in glass tubes containing food and monitored for their locomotor activity using the DAM system. Infrared beam crosses in 30 min bins were recorded. The circadian period and rhythmicity were estimated from locomotor activity data collected for 10 days in DD.

### Statistical analyses

All the statistical analyses were performed using IBM SPSS Statistics 22 (IBM Japan, Ltd.) or BellCurve for Excel (Social Survey Research Information Co., Ltd.) except for the comparisons of MI. The circadian period, and rhythmicity were estimated from the locomotor activity data collected for 10 days in DD. All statistical analyses except for the comparisons of MI, circadian period, and rhythmicity were performed as previously described [5,22,55].

In the statistical analysis of CI, when the data were not distributed normally, we carried out the arcsine and square root transformations of the data. When the basic data or transformed data were normally distributed, Student's $t$-test was used for comparisons. When the basic and transformed data were not distributed normally, we used the Mann–Whitney $U$ test for comparisons. A randomization test based on the bootstrapping method was used in the statistical analysis of MI with an open R script [57]. The free statistical package R was used for these tests.

In sleep analysis, when the basic or transformed data are distributed normally, one-way ANOVA followed by post-hoc analysis using Scheffe's test was carried out for multiple pairwise comparisons. For multiple group analysis of onparametric data, we used nonparametric ANOVA (Kruskal–Wallis test) followed by the rank-sum test for multiple pairwise comparisons.

In the statistical analysis of circadian rhythms, significant circadian rhythmicity was defined as the presence of a periodogram power peak that extends above the significance line ($P < 0.05$) in chi-square analysis. Clocklab software (Actimetrics) was used to analyze the circadian period and rhythmicity in each individual fly. All basic data of the circadian period were not normally distributed. The transformed data using arcsine transformation, log transformation, and square root transformation also did not show normal distribution. Thus, we used the Kruskal–Wallis test for comparisons of the circadian period. We used the $G$ test with William's correction to assess whether the percentage of rhythmicity (% rhythmic in Figs S4 and 7) differed significantly between the GAL4 control and the UAS control, or between the GAL4 control and the $F_1$ hybrids carrying both GAL4 and UAS.

In qRT-PCR, the mean (± SEM) ratio was calculated using data from five independent assays. Since the basic or log-transformed data were normally distributed, one-way ANOVA followed by post-hoc analysis using Scheffe's test was used.

## Supporting information

**S1 Fig. Real-time qRT-PCR analysis of *Tdc2* and *Tbh* mRNA expression levels.** (A–D) A pan-neural GAL4 line, *nSyb*-GAL4, was used to knockdown *Tdc2* or *Tbh* in the experiments. *nSyb*-GAL4/+ males were used as the control. Mean±SEM was calculated from five replicates. We visualized the data using a bar chart with individual data points (black circles). NS, not significant. $N=5$ in each bar. **, $P<0.01$; ***, $P<0.001$; NS, not significant. (A) UAS-*Tdc2* RNAi, (B) UAS-*Dicer2*; UAS-*Tdc2* RNAi, (C) UAS-*Tbh* RNAi #1, and (D) UAS-*Tbh* RNAi #2 were used in the experiments. (TIF)

**S2 Fig. *Tbh* knockdown does not impair STM.** (A) LNd-split-GAL4/UAS-*Tbh* RNAi #1 males were used in the experiments. LNd-split-GAL4/+ and UAS-*Tbh* RNAi #1/+ males were used as the control. (B)*Pdf*-GAL4/UAS-*Tbh* RNAi #1 males were used in the experiments. *Pdf*-GAL4/+ and UAS-*Tbh* RNAi #1/+ males were used as the control. (A, B) Males in each genotype were tested 1 h after 1 h conditioning. We visualized the data using a box plot with individual data points (white circles). See Fig 1 for an explanation of box plots. **, $P<0.01$; ***, $P<0.001$; $P$, probability; $U$, Mann–Whitney $U$; $N$, sample size in each box. (TIF)

**S3 Fig. *Tbh* knockdown in LNds has little effect on sleep.** All sleep/wake parameters (daily sleep pattern, total day sleep, total night sleep, sleep bout number, sleep bout duration, waking activity index) were analyzed using the data averaged over 3 days of LD. (A–F) Error bars show SEM in each figure. Black circles and bars, LNd-split-GAL4/+; gray circles and bars, UAS-*Tbh* RNAi #1/+; red circles and bars, LNd-split-GAL4/ UAS-*Tbh* RNAi #1. *, $P<0.05$; **, $P<0.01$; ***, $P<0.001$; NS, not significant. (A) Daily sleep patterns of control and experimental flies. (B) Total sleep amount during the day. (C) Total sleep amount during the night. (D) Sleep bout number during day and night. (E) Sleep bout duration during day and night. (F) Waking activity indices during day and night. (TIF)

**S4 Fig. *Tbh* knockdown in LNds has little effect on circadian rhythms.** Double-plotted actogram in DD. The bar above each actogram indicates subjective day (gray) and night (black). Each actogram shows locomotor activity for 10 days in DD at 25∘C. LNd-split-GAL4/UAS-*Tbh* RNAi #1 males were used. LNd-split-GAL4/+ and UAS-*Tbh* RNAi #1/+ males were used as the control. $N$, sample size. For the percentage of rhythmicity (% rhythmic), UAS-*Tbh* RNAi #1/+ or LNd-split-GAL4/UAS-*Tbh* RNAi #1 males were compared with GAL4 control males. We used the Kruskal–Wallis test for comparisons of the circadian period. The same letters in superscripts indicate no significant difference ($P>0.05$). $N$, sample size. (TIF)

**S5 Fig. Tbh-GFP co-expressing neurons in *R78G02*-positive neurons.** (A–C) UAS-*mCherry.NLS*/+; *R78G02*/*Tbh-GFP* males were used in the experiments. (A) Confocal section images at LNd level of the adult brain. Scale bars represent 20 µm. Arrows, LNds; Green, Tbh-GFP; Blue, mCherry.NLS; Magenta, ITP. (B) Confocal section images at 5th s-LNv level of the adult brain. Scale bars represent 20 µm. Arrow heads, 5th s-LNv; Green, Tbh-GFP; Blue, mCherry.NLS; Magenta, ITP. (C) Confocal images of posterior view of the adult brain. Scale bars represent 100 µm (left image) and 20 µm (six images on the right). Arrows, Tbh-GFP-positive non-clock neurons; blue, mCherry,NLS; green, Tbh-GFP. (TIF)

**S6 Fig. *Tbh* knockdown in Pdf neurons has little effect on sleep.** All sleep/wake parameters (daily sleep pattern, total day sleep, total night sleep, sleep bout number, sleep bout duration, waking activity index) were analyzed using the data averaged over 3 days of LD. (A–F) Error bars show SEM in each figure. Black circles and bars, *Pdf*-GAL4/+; gray circles and bars, UAS-*Tbh* RNAi #1/+; red circles and bars, *Pdf*-GAL4/UAS-*Tbh* RNAi #1. ***, *P*<0.001; NS, not significant. (A) Daily sleep patterns of control and experimental flies. (B) Total sleep amount during the day. (C) Total sleep amount during the night. (D) Sleep bout number during day and night. (E) Sleep bout duration during day and night. (F) Waking activity indices during day and night.
(TIF)

**S1 Data. Excel spreadsheet containing the underlying numerical data for Figs 1D, 2D, 2E, 2F, 2G, 3A, 3B, 3C, 3D, 4C, 4D, 5E, 5F, 5G, 6A, 6B, 6C, 7A, 7B, 7C, 7D, 7E, 7F, 7G,S1A, S1B, S1C, S1D, S2A, S2B, S3A, S3B, S3C, S3D, S3E, S3F, S4, S6A, S6B, S6C, S6D, S6E, and S6F.**
(.XLSX)

## Acknowledgments

We thank Sayuri Takakura and Emiko Nakagawa for technical assistance and Shoma Sato and Show Inami for providing critical comments. We are grateful to the Bloomington Drosophila Stock Center and the Drosophila Genomics and Genetic Resources (DGGR, Kyoto Stock Center) for providing the fly strains.

## Author contributions

**Conceptualization:** Yuto Kurata, Takaomi Sakai.

**Data curation:** Yuto Kurata.

**Formal analysis:** Yuto Kurata.

**Funding acquisition:** Takaomi Sakai.

**Investigation:** Yuto Kurata.

**Methodology:** Yuto Kurata, Taishi Yoshii.

**Project administration:** Takaomi Sakai.

**Supervision:** Takaomi Sakai.

**Validation:** Yuto Kurata.

**Visualization:** Yuto Kurata.

**Writing – original draft:** Yuto Kurata, Taishi Yoshii, Takaomi Sakai.

**Writing – review & editing:** Yuto Kurata, Takaomi Sakai.

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
