## [Decision Letter · Decision Letter 0]

26 Nov 2025

PGENETICS-D-25-01073

Octopamine signaling from clock neurons plays dual roles in Drosophila long-term memory

PLOS Genetics

Dear Dr. Sakai,

Thank you for submitting your manuscript to PLOS Genetics. After careful consideration, we feel that it has merit but does not fully meet PLOS Genetics's publication criteria as it currently stands. Therefore, we invite you to submit a revised version of the manuscript that addresses the points raised during the review process.

Please submit your revised manuscript within by Dec 26 2025 11:59PM. If you will need more time than this to complete your revisions, please reply to this message or contact the journal office at plosgenetics@plos.org. Please include the following items when submitting your revised manuscript:

We look forward to receiving your revised manuscript.

Kind regards,

Giovanni Bosco, Ph.D.

Section Editor

PLOS Genetics

Giovanni Bosco

Section Editor

PLOS Genetics

Aimée Dudley

Editor-in-Chief

PLOS Genetics

Anne Goriely

Editor-in-Chief

PLOS Genetics

**Journal Requirements:**

3) We note that your Data Availability Statement is currently as follows: "All data supporting the findings are provided within the manuscript and its supporting information files.". Please confirm at this time whether or not your submission contains all raw data required to replicate the results of your study. Authors must share the “minimal data set” for their submission. PLOS defines the minimal data set to consist of the data required to replicate all study findings reported in the article, as well as related metadata and methods (https://journals.plos.org/plosone/s/data-availability#loc-minimal-data-set-definition).

4) Please amend your detailed Financial Disclosure statement. This is published with the article. It must therefore be completed in full sentences and contain the exact wording you wish to be published.

1) State what role the funders took in the study. If the funders had no role in your study, please state: "The funders had no role in design, data collection and analysis, decision to publish, or preparation of the manuscript.".

**Reviewers' comments:**

Reviewer's Responses to Questions

**Comments to the Authors:**

**Please note that one review is uploaded as an attachment.**

Reviewer #1: The authors described the role of octopamine from two distinct populations of clock neurons on long-term memory (LTM), which is a non-clock function in Drosphila melanogaster. They found PDF-neurons, which includes large and small ventral lateral neurons (sLNvs and lLNvs) functions in the consolidation of LTM, while dorsal lateral neurons (LNd) functions in the maintenance of LTM. Although these neurons functions in circadian rhythm regulation and octopamine in sleep regulations, alteration of octopamine signals from these neurons had no effects of circadian rhythm and sleep. The experiments were well designed and have been performed under contemporary standards, and the results well supported their conclusion. This study described non-clock function of clock neurons, which is novel and brought significant insights into this field with its clarification of the molecular mechanism.

I have a couple of minor comments as follows, before it can be accepted;

1. It is preferable to show the RNAi specificity. The authors used only 1 RNAi line in PDF neuron experiments, which can be improved if they used additional lines, like LNd. In addition, rescue experiments will also strengthen the conclusion.

2. As for R78G02 experiments, the authors guessed the involvement of non-clock neurons. This conclusion solely depends on the results of R54D11 and LNd-split GAL4 experiments. However, the efficiencies of RNAi by different GAL4 are different, even they are expressed in the same cells. Therefore, the results can be interpreted in another way. It is also preferable to use more specific GAL4 drivers, which show positive effects in memory functions.

3. The authors described that the baseline of the courtship index (naïve CI) were different between LNd-split-GAL4/UAS-Tbh RNAi “1 and Pdf-GAL4/UAS-Tbh RNAi #1. Since memory index of LTM is affected by the native CI, they should mention how this difference may affect the results in the discussion.

4. The authors concluded Tbh knockdown did not affect the circadian rhythm and sleep. But in the supplemental figures (S2, S6), there are minor changes in a couple of parameters. Thus, they should rephrase, maybe “There were minor effects which are not likely to affect LTM.”

Reviewer #2: uploaded

**Have all data underlying the figures and results presented in the manuscript been provided?**

Reviewer #1: Yes

Reviewer #2: Yes

PLOS authors have the option to publish the peer review history of their article (what does this mean? ). If published, this will include your full peer review and any attached files.

**Do you want your identity to be public for this peer review?** For information about this choice, including consent withdrawal, please see our Privacy Policy .

Reviewer #1: No

Reviewer #2: No

**Figure resubmission:**
---

## [Decision Letter · Decision Letter 1]

3 Feb 2026

Dear Dr Sakai,

We are pleased to inform you that your manuscript entitled "Octopamine signaling from clock neurons plays dual roles in Drosophila long-term memory" has been editorially accepted for publication in PLOS Genetics. Congratulations!

Yours sincerely,

Giovanni Bosco, Ph.D.

Section Editor

PLOS Genetics

Giovanni Bosco

Section Editor

PLOS Genetics

Aimée Dudley

Editor-in-Chief

PLOS Genetics

Anne Goriely

Editor-in-Chief

PLOS Genetics

BlueSky: @plos.bsky.social

Comments from the reviewers (if applicable):

Reviewer's Responses to Questions

**Comments to the Authors:**

Reviewer #1: The authors responded to all my concerns satisfactory, and I have no more comments.

Reviewer #2: I am satisfied with the additions and changes that have been made.

**Have all data underlying the figures and results presented in the manuscript been provided?**

Reviewer #1: Yes

Reviewer #2: Yes

PLOS authors have the option to publish the peer review history of their article (what does this mean? ). If published, this will include your full peer review and any attached files.

**Do you want your identity to be public for this peer review?** For information about this choice, including consent withdrawal, please see our Privacy Policy .

Reviewer #1: **Yes:** Kazuhiko Kume

Reviewer #2: No

**Data Deposition**

http://datadryad.org/submit?journalID=pgenetics&manu=PGENETICS-D-25-01073R1

**Press Queries**

---

## [Editor Report · Acceptance letter]

PGENETICS-D-25-01073R1

Octopamine signaling from clock neurons plays dual roles in Drosophila long-term memory

Dear Dr Sakai,

We are pleased to inform you that your manuscript entitled "

Octopamine signaling from clock neurons plays dual roles in Drosophila long-term memory" has been formally accepted for publication in PLOS Genetics! Your manuscript is now with our production department and you will be notified of the publication date in due course.

With kind regards,

Lilla Horvath

PLOS Genetics

On behalf of:
